# Synthesis of (1,10-Phenanthroline-*κ*^2^*N*,*N*′)(β^2^-Methyl- and β^2^-PhenylAlaninate-*κ*^2^*N*,*O*)Copper(II) Nitrate Complexes and Their Antiproliferative Activity on MCF-7 and A549 Cancer Cell Lines

**DOI:** 10.3390/molecules30030634

**Published:** 2025-01-31

**Authors:** Leticia Chavelas-Hernández, Luis G. Hernández-Vázquez, Jonathan R. Valdez-Camacho, Adrián Espinoza-Guillén, Carlos A. Tavira-Montalvan, Angélica Meneses-Acosta, Eusebio Juaristi, Lena Ruiz-Azuara, Jaime Escalante

**Affiliations:** 1Centro de Investigaciones Químicas-IICBA, Universidad Autónoma del Estado de Morelos, Av. Universidad 1001, Cuernavaca 62209, Mexico; letychh@gmail.com (L.C.-H.); valca@uaem.mx (J.R.V.-C.); 2Departamento de Química, Universidad Autónoma Metropolitana-Iztapalapa, San Rafael Atlixco 186, Col. Vicentina, Iztapalapa, Ciudad de Mexico 09340, Mexico; 3Departamento de Química Inorgánica y Nuclear, Facultad de Química, Universidad Nacional Autónoma de México, Av. Universidad 3000, Ciudad de Mexico 04510, Mexico; adrianeg@quimica.unam.mx (A.E.-G.); lenar701@gmail.com (L.R.-A.); 4Facultad de Farmacia, Universidad Autónoma del Estado de Morelos, Av. Universidad 1001, Cuernavaca 62209, Mexico; carlos.taviramon@uaem.edu.mx (C.A.T.-M.); angelica_meneses@uaem.mx (A.M.-A.); 5Departamento de Química, Centro de Investigación y de Estudios Avanzados, Avenida I.P.N. 2508, Ciudad de Mexico 07360, Mexico; juaristi@relaq.mx; 6El Colegio Nacional, Calle Luis González Obregón 23, Centro Histórico, Ciudad de Mexico 06020, Mexico

**Keywords:** β^2^-aminoacidates, Casiopeínas^®^, copper, cytotoxic activity

## Abstract

In recent years, metallodrugs have been playing an important role, showing to be more efficient in the treatment of several diseases, such as cancer. Indeed, it is important to synthesize novel molecules to be used as more effective agents against cancer. In the present paper, the synthesis of two new molecules belonging to Casiopeínas^®^ is reported. These compounds present a β^2^-aminoacidate derivative as the secondary ligand. The novel metal complexes were characterized by high-resolution mass spectrometry, FT-IR, UV-Vis, EPR, effective magnetic moment and cyclic voltammetry measurements, and single crystal X-ray diffraction analysis. Furthermore, these compounds were evaluated in vitro against the cancer lines MCF-7 (breast cancer) and A549 (lung cancer).

## 1. Introduction

Nowadays, cancer constitutes a major health concern worldwide. According to estimated data for the year 2024 [1], just in the USA, are anticipated around 2 million new cancer cases, with an estimated 612,000 deaths. The predominant types of cancer responsible for such mortality in people are pancreatic, prostate, liver, kidney, colorectal, and breast. In this regard, chemical research directed to finding new organic or inorganic drugs aiming to increase the survival rates of cancer patients and their quality of life has developed cisplatin, which has been on the World Health Organization’s list of essential medicines since 2015.

Throughout human history, metals have played an important role, not only because they are components in objects essential to our day-to-day activities, but also because they have been used for medicinal purposes in civilizations such as the Chinese [2], Greek [3], and Egyptian [4]. In 1907, Alfred Bertheim synthesized compound 606, diamino dihydroxy arsenobenzol, best known as arsphenamine, which proved to be effective for the treatment of syphilis [5]. Half a century later, in the year 1965, Barnett Rosenberg, during an investigation of the possible effects of electric fields on bacterial growth processes, discovered a platinum-based compound that inhibited cell division [6]. Initially, this compound was identified as [PtCl_4_(NH_3_)_2_], and later, as *cis*-[PtCl_2_(NH_3_)_2_] (cisplatin) [7]. Several years later, the FDA approved cisplatin for the treatment of testicular cancer.

Currently, cisplatin is one of the best-selling drugs worldwide for cancer treatment. Furthermore, a new generation of platinum-containing drugs has been developed, providing cancer patients worldwide with new, effective treatments [8]. Moreover, the success achieved by cisplatin motivated the use of other transition metals and various heterometals in the preparation of relevant metallodrugs [9,10,11,12,13,14,15,16,17,18,19,20]. Nevertheless, in addition to the side effects caused by metallodrugs, including nausea, vomiting, myelosuppression [21], nephrotoxicity [22], ototoxicity [22], and neurotoxicity [21,23], most of them present serious drawbacks such as intrinsic or acquired resistance. Therefore, there is a continuous need for the design and synthesis of new drugs to avoid these limitations and to minimize side effects. 

Metallic complexes, especially those containing copper, frequently exhibit biological activity [24,25,26,27]. It is also known that copper is a trace element required for some enzymes that plays an important role [28]. In this context, Casiopeínas^®^, which was synthesized for the first time in 1987, emerged as an alternative to minimize or eliminate the drawbacks associated with most metal complexes. Its general formula is denoted by [Cu(N–N)(O–O)]NO_3_ or [Cu(N–N)(N–O)]NO_3_, where (N–N) represents a primary ligand, typically a 2,2′-bipyridine or 1,10-phenanthroline derivative. On the other hand, (O–O) corresponds to acetylacetonate or salicylaldehydate derivatives, and (N–O) can be aminoacidate or peptidate moieties as secondary ligands [29,30].

On the other hand, some β-amino acids exhibit biological activity on their own [31,32], or they are incorporated in structures such as β-peptides, cyclo-β-peptides, β-lactams, and other biologically active compounds [30,31,32,33,34,35,36]. In this regard, β-amino acids can be classified according to their substitution pattern as β^2^, β^3^, or β^2,3^ [29,37]. Of special interest are β^2^-amino acids such as 3-amino-2-methylpropanoic acid because of their biological and chemical properties [36,37,38,39,40]. This β-amino acid is particularly interesting because it is the only one present in both enantiomeric forms in the human body: the (*R*) enantiomer is a thymine catabolite and is present in the urine [41], while the (*S*) enantiomer is a valine catabolite and is present in plasma [42]. Finally (*R*)-3-amino-2-methylpropanoic acid is a β-amino acid residue present in Cryptophycin A, which is a potent antitumoral [43].

Recently, we synthesized three Cu(II) mixed complexes incorporating 1,10-phenanthroline as the primary ligand and β-alaninate or β^3^-aminoacidate derivatives as secondary ligands. Those Cu(II) complexes were characterized by spectroscopic and electroanalytical techniques, showing a square-based pyramidal geometry as the result of solvent coordination. In vitro studies revealed that those complexes exhibit growth inhibition activity against the MCF-7 and A549 tumor cell lines [16]. These cell lines are common models used for in vitro studies of breast and lung cancer, respectively. The present paper describes the synthesis and characterization of two new Cu(II) complexes employing 1,10-phenanthroline as the primary ligand and β^2^-aminoacidates as the secondary ligand. The cytotoxic activity of these novel complexes was also evaluated against the MCF-7 and A549 cell lines.

## 2. Results and Discussion

### 2.1. Synthesis and FT-IR Characterization

Despite the enormous interest in the chemistry of α-amino acids, only a few reports of Casiopeínas^®^ containing α-amino acids have been reported [44,45]. Moreover, even fewer examples of analogous complexes incorporating β-amino acids have been published [16,46]. Previously, the effect of a substituent at the β^3^ position on the antiproliferative activity on MCF-7 (breast cancer) and A549 (lung cancer) tumor cell lines was explored. The β^3^-amino acid complexes exhibited similar activity to other Casiopeínas^®^ that had been reported previously [47]. In order to explore the potential effect of substituents at the β^2^ position, the secondary ligands (*rac*)-3-amino-2-methylpropanoic acid (**1**) and (*rac*)-3-amino-2-phenylpropanoic acid (**2**) were incorporated. Ligand **1** (L1) was synthesized by a typical benzylamine Michael addition to methacrylate derivative followed by debenzylation and hydrolysis (Figure 1A), while ligand **2** (L2) was synthesized by a modified malonic acid synthesis to obtain an α-β-unsaturated derivative followed by benzylamine Michael addition, debenzylation, and hydrolysis (Figure 1B) [48,49]. Complexes were synthesized as described using stoichiometric amounts of ligands (Figure 1C) [16].

Cu(II) complexes show characteristic IR absorption bands in the 3200–3600 cm^−1^ range, corresponding to ν(N-H) and ν(O-H) stretching vibrations in the β-aminoacidate ligand and coordinating solvent. The carboxylate group’s stretching vibrations are present in the 1584–1588 cm^−1^ range, assigned to C=O stretching, and the 1224–1234 cm^−1^ range for stretching O–C vibration. The C–H bending bands at 721–723 cm^−1^ and 1511–1519 cm^−1^ stretching C=N band confirm the presence of the phenanthroline ligand. Finally, characteristic nitrate absorption bands can be found in the 1354–1380 cm^−1^ range (Table 1).

### 2.2. Single-Crystal X-Ray Diffraction Analysis

Suitable crystals of complexes **3** and **4** were grown from undisturbed MeOH/Et_2_O or EtOH/Et_2_O solutions at room temperature. Aqua(1,10-phenanthroline-κ^2^*N*,*N*’)(β^2^-methylalaninate-κ^2^*N*,*O*)copper(II) nitrate ([**Cu(H_2_O)(phen)L1**]**NO_3_·2H_2_O**) (**3**) crystallizes in the monoclinic space group *C*2/*c*. The Cu(II) ion exhibits a five-coordinated behavior in a square-based pyramidal geometry with 1,10-phenanthroline and β^2^-methylalaninate ligands as the base and a water molecule in the apical position (Figure 1).

On the other side, ethanol(1,10-phenanthroline-κ^2^*N,N*’)(β^2^-phenylalaninate-κ^2^*N,O*)copper(II) nitrate ([**Cu(EtOH)(phen)L2**]**NO_3_**) (**4**) crystallizes in the monoclinic space group *P*2_1/*c*_. The Cu(II) ion also exhibits a five-coordinated behavior in a square-based pyramidal geometry, but with an ethanol molecule in the apical position (Figure 2).

The observed difference in the coordinating behavior of the solvent molecule in **3** and **4** might be a consequence of crystal lattice packing effects. As stated by Halcrow [52], an axial distance Cu-L of up to 2.4 Å might be consistent with the existence of a bond giving rise to the square-based pyramid geometry. The X-ray structures of complexes obtained are also consistent with related five-coordinated Cu(II) Casiopeínas^®^ reported previously [16,53]. Selected bond distances and angles for Cu(II) complexes **3** and **4** are listed in Table 2.

### 2.3. Crystalline Structure and Hirshfeld Surface Analysis

The crystalline structure of ([Cu(H_2_O)(phen)L1]NO_3_·2H_2_O) **3** is stabilized by a network of hydrogen bonding interactions from N/O-donor and -acceptor groups of nitrate, water, and β^2^-methylalaninate (Figure 3). The linkage between molecules goes through coordination water and nitrate molecules, viz., Cu–OH····O_nitrate_, O_nitrate_····H–N_β2-methylalaninate_, and Cu–OH····OCO. In Table 3 are summarized the H–bond distances and angles for compound **3** and **4** as calculated by PLATON software 2023.1 [54].

On the other hand, the ([Cu(EtOH)(phen)L2]NO_3_) **4** crystalline network is even more stabilized because of hydrogen interactions from N/O-donor and -acceptor groups such as nitrate, ethanol, and β^2^-phenylalaninate, but it also presents van der Waals interactions between aromatic groups and the coordination ethanol (Figure 4).

Hirshfeld surface analysis provides intermolecular interactions in the crystal structure, which can be viewed through 3D molecular surfaces and 2D fingerprints. These interactions are color-coded represented by fingerprints and contour surfaces. In Figure 5 are depicted the d_norm_ surfaces for compounds **3** and **4**, where the circular red areas indicate intermolecular interactions [55], blue areas are regions distant for atom interactions [56], and white areas around phenanthroline ligands represent the π-stacking interactions.

Fingerprints plots are 2D Hirshfeld surfaces that can be organized to highlight contributions to the crystal packing of specific atom–atom interactions [57]. As can be seen, for fingerprints of both compounds (Figure 6 and Figure 7), C····H interactions are more prominent in compound **4**, which is indicative of more van der Waals interactions than in compound **3**. On the other hand, O····H interactions are lower than in compound **3**, around a 5% difference, because of the change in coordination solvent. As stated before, in compound **3,** coordination water serves as a bridge for the linkage between molecules, while in compound **4**, coordination ethanol serves this purpose through van der Waals interactions with aromatic rings and hydrogen bondings with carbonyl groups.

### 2.4. UV–Visible Spectroscopy

Measurements were made by triplicate in water. Both Cu(II) complexes show an absorption band at λ_max_ = 272 nm, which is attributed to π→π* transitions derived from the phenanthroline ligand. Furthermore, absorption bands at λ_max_ = 697 nm for complex **3** and λ_max_ = 652 nm for complex **4** are due to the d-d transitions d_xz_, d_yz_→ dx^2^ − y^2^ [58]. These are not evident at low concentrations (an overlay in region 400–800 nm at 1 × 10^−3^ M concentration is shown) (Figure 8A,B). Meanwhile, in the solid state, they are more evident (Figure 8C,D) (see Appendix A).

### 2.5. Electron Paramagnetic Resonance Spectroscopy

EPR spectroscopy can distinguish ground states on the basis of the **g** tensor in the anisotropic spectra. In coordination geometries such as the elongated octahedron, square planar, or square-based pyramid, the ground state is dx^2^ − y^2^; while in a compressed octahedron or trigonal bipyramid, the ground state is dz^2^. The **g** value is an empirical parameter and is affected by changes in the electronic structure. If dx^2^ − y^2^ is the ground state, a relation g_z_ > g_x_ = g_y_ is expected; on the other hand, if dz^2^ is the ground state, g_x_ = g_y_ > g_z_ is expected [59].

EPR spectra for both Cu(II) complexes were obtained at 77 K in frozen MeOH (Figure 9A,B), and the spin parameters are summarized in Table 4. EPR spectra show an axial symmetry, and it can be seen that g_z_ > g_x_ = g_y_, suggesting a ground state in dx^2^ − y^2^; therefore, a square-based pyramid geometry is expected for both Cu(II) complexes, and g_z_ < 2.3 values indicate a covalent behavior, while g_z_ > 2.3 values are indicative of ionic behavior. Also, g_z_/A_z_ ratio values between 130 and 150 indicate slight to moderate distortion in the Cu-L bonds [60]. Another way to determine the Cu(II) center geometry by using the spectrum at the X-band is plotting A_z_ versus g_z_ and then finding that coordinate in the Peisach–Blumberg relationships. These correlations are imprecise because of the overlapping of 4N, 2N_2_O, and to a lesser degree, 4O systems, but they can provide information about the coordination of sulfur or oxygen in the absence of electron-nuclear superhyperfine interactions [61].

### 2.6. Cyclic Voltammetry and Conductometric Measurements

Half-wave potentials (E_1/2_, Figure 10) were determined as the mean of seven measurements at different scan rates by the half-sum of the cathodic and anodic peak potentials. Voltammograms are in agreement with a single-electron transfer process, which corresponds to the redox couple Cu(II)/Cu(I). Since no other peaks are visible, neither reduction nor oxidation of ligands takes place, suggesting stability to degradation. Electrochemical behavior for both complexes is similar; the cathodic peak is more evident in complex **4** due to phenyl substituent, and ΔE_p_ between E_pc_ and E_pa_ is greater than 60 mV, which suggests a quasi-reversible behavior since peak separation varies with scan rate [62,63].

In single-crystal X-ray diffraction, the ion nitrate is observed as a counter ion for Cu(II) complexes; thus, a 1:1 electrolyte solution is expected when preparing aqueous solutions. Conductivity measurements of **3** and **4** in water were performed, giving a molar conductivity of 124.25 S·cm^2^/mol and 129.41 S·cm^2^/mol, respectively, which is an expected value for these solutions.

### 2.7. Effect on Growth Inhibition (IC_50_)

The effect of Cu(II) complexes on cell growth inhibition of A549 (lung cancer) and MCF-7 (breast cancer) tumor cell lines compared with the non-cancerogenous cell lines HaCaT (epithelial origin) and HEK293 (fibroblastic origin) was determined by the IC_50_ at 24 h and is summarized in Table 5.

Such analysis in terms of IC_50_ is presented in Table 5 and demonstrates that complexes **3** and **4** are significantly more potent than cisplatin and Cu(NO_3_)_2_ for all the cell lines tested. Their low IC_50_ values (<1 µM) reflect high efficacy, making them potential complexes with anticancer activity. In contrast, Cu(NO_3_)_2_ exhibits high IC_50_ values, suggesting moderate cytotoxic efficacy. The classification of cytotoxicity according to IC_50_ values is widely reported in the literature [64]. Thus, we can confirm the active species are those from crystalline structures. As can be seen in the Appendix A, both complexes might be suffering from degradation at long times, but at experimental times, there is enough concentration of both to show the cytotoxic effect.

Compared with Casiopeínas^®^ with β^3^-aminoacidate ligand, which had an IC_50_ in the range 12.89–36.85 µM [16], these Casiopeínas^®^ with β^2^-aminoacidate ligands exhibit an activity more than 80 times higher at the same concentrations.

The selectivity of the compounds toward both tumor lines (A549 and MCF-7) was determined by the Selectivity Index (SI). It showed that compound **3** presents low specificity, with values of 0.875 and 0.6, respectively, for A549 and MCF-7. On the other hand, compound **4** shows moderate selectivity toward the A549 tumor line (SI = 1.59), but its specificity is low against MCF-7 (SI = 0.87). This suggests that compound **4** has limited potential, being more promising for the A549 tumor line.

Although the compounds exhibit low SI, the results obtained allow addressing a chemical optimization approach to improve selectivity, serving as a starting point for the design of more specific analogues, e.g., produg design [65]. In addition, advanced formulation strategies, such as nanoparticles or liposomal vehicles, allow targeting compounds to solid tumors, reducing their toxicity in normal cells [66], or strategies involving combinations with other drugs or modification of the therapeutic window can be implemented [67].

#### 2.7.1. Staining with Fluorescent Dyes Acridine Orange (AO) and Propidium Iodide (PI)

The results of acridine orange and propidium iodide (AO/IP) staining on A549, MCF-7, HEK293, and HaCaT cell cultures exposed to cisplatin and the different complexes **3** and **4** are shown in Figure 11. Control (untreated) cells show a defined reported structure without evident morphological changes, and no internalization of the IP dye into the cell is observed, indicating no compromise of cell membrane integrity. Cells treated with cisplatin [25 µM] show red nuclei, indicating a decrease in cell viability; the morphology is consistent with a programmed cell death process (DNA condensation and presence of intracellular vacuolization). Cultures treated with complex **3** present a morphology consistent with a programmed cell death process. The A549 cells show bright spots in the cytoplasm, indicating high lysosomal activity and nuclear fragmentation, and PI was internalized in some cells, staining the DNA. The cytotoxic effect of complex **3** is more evident in MCF-7 cells, where a greater effect on cell adhesion and programmed cell death characteristics is observed: rounded cells, loss of cytoplasmic structure, and nuclear fragmentation. It is shown that the culture was in a process of early apoptosis and in late apoptosis, where there is complete staining of the nucleus by the IP dye. Also, extensions of the plasma membrane are separated from the cell, containing cytoplasm and cell fragments (blebbing). HEK293 cells show loss of adhesion, nuclear fragmentation, and loss of cytoplasmic structure. HaCaT cells show high lysosomal activity in the cytoplasm, chromatin condensation, and no compromise of cell adhesion as in the case of MCF-7 cells. The morphological changes observed in the cultures exposed to complex **3** are presented in the same way as the cultures with complex **4**. Likewise, it is observed that the MCF-7 cell cultures are more susceptible to complex **4,** and the HaCaT cell cultures are the least susceptible to cell adhesion, even though they present morphological characteristics typical of an early cell death process.

Epifluorescence microscopy (Nikon Eclipse E600, Nikon Corporation. Tokio, Japan) with OA/IP staining indicated that complexes **3** and **4** induced morphological features that, in principle, indicate a cell death process in cultures of A549, MCF-7, HEK293, and HaCaT cells. The described morphologies of A549, MCF-7, HEK293, and HaCaT cell cultures exposed to cisplatin corresponds to those reported in the literature; cisplatin induces apoptosis in cancer and non-cancer cell cultures [68]. This apoptotic morphology reported with AO/IP staining corresponds to changes in cell shape; loss of cell adhesion; formation of apoptotic bodies; alterations in the nucleus, including chromatin condensation and nuclear fragmentation; vacuolization of the cytoplasm due to cell stress; and changes in the organization of the cytoskeleton [69,70,71].

Comparing the effect of cisplatin as a positive control of programmed cell death with complexes **3** and **4**, it was observed that both can induce cell death comparable to the effects reported for cisplatin. It is important to mention that, although the effect of complexes **3** and **4** is consistent with programmed cell death, each cell line responds differently to the compounds evaluated. For example, A549 and HaCaT cells present high lysosomal activity, which is closely related to cell death by apoptosis [72]. MCF-7 cells do not show lysosomal activity, but formation of apoptotic bodies and blebbing are observed; the presence of blebbing is one of the morphological and biochemical changes that occur during apoptosis, ensuring an orderly and regulated elimination of apoptotic cells [73]. For all cultures, it was observed that complexes **3** and **4** induce an effect on the nuclear structure, presenting chromatin condensation, contributing to the fragmentation of the nucleus and subsequent controlled dismantling of the apoptotic cell [74].

#### 2.7.2. Nuclei Staining with Hoechst 33258

The results of nuclei staining with Hoechst 33258 of the A549, MCF-7, HEK293, and HaCaT cultured cells exposed to cisplatin and complexes **3** and **4** are shown in Figure 12. This staining is based on the affinity of the Hoechst 33258 molecule for DNA, which allows visualization of the cell nuclei. Control cells (untreated) show defined oval-shaped nuclei, where no evidence of nuclear changes is observed. In A549 and HEK293 cells, chromatin condensation is observed in some nuclei due to the fact that in these control cultures there are cells in the process of cell division. Cultures treated with cisplatin [25 µM] show an evident change in nuclear structure; the increase in fluorescence intensity with respect to the untreated cultures is due to chromatin condensation as a response to cell damage caused by the presence of cisplatin, inducing a programmed cell death process such as apoptosis. During a cell death process such as apoptosis, the nuclear morphology undergoes major changes such as chromatin condensation, fragmentation of the nucleus, and formation of nuclear vesicles [75].

The capacity of cisplatin to induce apoptosis in cancer cells is widely reported. This drug induces damage to DNA, provoking chromatin condensation and its fragmentation. Another effect is the depolarization and change in membrane potential, culminating in cell death by apoptosis [76,77]. For A549, MCF-7, HEK293, and HaCaT cell cultures treated with complexes **3** and **4**, at concentrations corresponding to the IC_50_ shown in Table 5, staining with Hoechst 33258 allows visualizing a decrease in the size of the nuclei, irreversible chromatin condensation, and nuclear fragmentation. The compromise of cell adhesion is evident, indicating that the potency of these compounds is higher than that observed with cisplatin. Considering the concentrations used to determine the IC_50_ for complexes **3** and **4**, there is an increase of 83 times in the potency than in the case of cisplatin at such concentrations.

#### 2.7.3. DNA Fragmentation Assay

DNA fragmentation assay is a widely used technique to detect programmed cell death such as apoptosis in cells treated with potential anticancer compounds. During apoptosis, DNA fragmentation is a characteristic event that occurs in a specific manner in fragments of size multiples of 180–200 base pairs (bp). During these programmed cell death processes, effector caspases such as caspase-3, -6, and -7 are activated. These caspases cleave ICAD (Inhibitor of CAD), releasing CAD (caspase-activated DNase), which enters the nucleus and cleaves DNA between nucleosomes. CAD cuts the DNA into fragments of approximately 180–200 bp and multiples thereof, creating a “ladder” pattern on an agarose gel [78,79].

The visualization of the fragmentation pattern of DNA extracted from A549, MCF-7, HEK293, and HaCaT cell cultures is shown in Figure 13. Figure 13A, in lanes 2–5, shows DNA extracted from untreated cultures and high-molecular-weight DNA from viable cultures with no evidence of endonuclease activity. On the other hand, in lanes 6–9, treated with cisplatin [25 µM], DNA fragmentation is observed, and the size of the bands corresponds to multiples of 180 bp due to the activity of caspase-activated kinases (CADs). Cisplatin is considered a control of apoptosis in cellular assays due to its specific and well-defined mechanism of action related to DNA damage and activation of intrinsic and extrinsic apoptosis pathways, providing a robust control to compare with other experimental conditions [80,81].

Cultures treated with complex **3** (lanes 2–5) and complex **4** (lanes 6–9) are shown in Figure 13B. The concentrations correspond to the IC_50_s calculated previously (Table 5). The DNA fragmentation observed on the agarose gel is a specific indication of an apoptotic process induced by complexes **3** and **4**, as the “ladder” pattern of DNA fragments in multiples of 180–200 bp reflects the activation of effector caspases and caspase-activated DNase, confirming programmed cell death. The combination of acridine orange staining, Hoechst staining, and DNA fragmentation assay demonstrates that complexes **3** and **4** can promote programmed cell death in cultured A549, MCF-7, HEK293, and HaCaT cells. Each technique offers a different perspective on the apoptotic process. Acridine orange/propidium iodide staining morphologically differentiates living cells from apoptotic cells. Hoechst staining visualizes apoptosis-specific nuclear morphology. DNA fragmentation assay confirms the specific caspase-induced DNA degradation characteristic of apoptosis [82,83].

#### 2.7.4. Apoptosis Detection with Annexin V FITC and Propidium Iodide

Cytotoxic activity and the effects of complexes **3** and **4** on the induction of apoptosis in A549, MCF7, HEK293, and HaCaT cell cultures is shown in Figure 14 using cisplatin as a positive control, whose capacity to generate apoptosis is widely documented in the literature [84], and untreated cells as a negative control. In the untreated control group, no fluorescent signal is observed in the channels corresponding to FITC (green) and TRITC (red), indicating that the cells remain viable and do not show phosphatidylserine exposure or membrane permeability, characteristic of apoptosis or necrosis, respectively. In contrast, cisplatin treatment generates an intense signal in the green channel (FITC), corresponding to annexin V binding to exposed phosphatidylserine in apoptotic cells, and a signal in the red channel (TRITC), associated with necrosis or apoptosis in advanced stages.

Cultures treated with complexes **3** and **4** show a fluorescence pattern similar to that observed with cisplatin, although with variations in intensity and the number of cells affected, depending on the cell line. For all cell lines evaluated, a predominant signal is observed in the green channel (FITC), indicating a significant induction of apoptosis, while a fraction of cells presents a signal in the red channel (TRITC), suggesting the presence of late apoptosis or necrosis. The results demonstrate that complexes **3** and **4** have the ability to induce apoptosis efficiently, with no selectivity observed, showing an effect comparable to cisplatin. The absence of fluorescence in the negative control group validates the specificity of the signals observed in the treated groups, confirming that the apoptotic activity is a direct consequence of the action of the complexes.

This result, added to previous assays of cell viability, morphological changes (assessed by AO/IP and Hoechst dyes), and DNA fragmentation, reinforces the ability of complexes **3** and **4** to induce apoptosis in cells of tumor origin. The consistency of the results obtained with the different methodological approaches suggests that both complexes not only affect cell membrane integrity, as evidenced by phosphatidylserine exposure, but also induce nuclear changes characteristic of apoptosis, such as DNA condensation and fragmentation. This evidence becomes relevant in the context of the search for new drugs with anticancer activity, since the ability to trigger apoptosis efficiently in tumor cells is a key criterion for the development of therapeutic agents [85].

## 3. Materials and Methods

### 3.1. Reagents, Solvents, and Cell Lines

Copper(II) nitrate [Cu(NO_3_)_2_·H_2_O, Aldrich], KBr (Aldrich, St. Louis, MO, USA), 1,10-phenanthroline (Merck, Darmstadt, Germany) were acquired from commercial suppliers and used without further purification. MCF-7 (breast cancer) and A549 (lung cancer) cancer cell lines as well as HEK293 (fibroblastic origin) and HaCaT (epithelial origin) cells were purchased from the American Type Culture Collection (ATCC, Manassas, VA, USA).

### 3.2. Instrumental

#### 3.2.1. FT-IR

FT-IR spectra were obtained in a NICOLET 6700 ThermoScientific spectrophotometer (Waltham, MA, USA) in 400–4000 cm^−1^ range at 25 °C using ATR technique.

#### 3.2.2. Mass Spectrometry

Data were obtained in a mass spectrometer ESI-MS Bruker Daltonics—micrOTOF II (Billerica, MA, USA) equipment in positive polarity with mass detection in full scan mode in range 50–3000 *m*/*z*. Capillary voltage was set at 4500 V with an end plate offset −500 V. Nebulization set at 0.5 Bar, drying gas flow (N_2_) was set at 4.0 L/min, and temperature at 150 °C. Sample amounts of 0.5 mg were dissolved in 500 µL MeOH and then diluted 1:1000 in MeOH/formic acid (0.1%).

#### 3.2.3. UV-Vis

UV-Vis spectra were obtained at 25 °C in a Genesis 10S equipment (ThermoScientific) or an Agilent Cary 50 Spectrophotometer equipment (Santa Clara, CA, USA) with a 200–800 nm spectral window with 1 nm resolution in a 1 cm optical pass cell and 2 mL volume in MeOH and water. Samples used were 1 × 10^−3^, 2 × 10^−4^, 1.5 × 10^−4^, 1.125 × 10^−4^, 8.44 × 10^−5^, 6.33 × 10^−5^, 4.75 × 10^−5^, 2.37 × 10^−5^, and 1.18 × 10^−5^ M and were prepared by dilution in used solvent. Measurements were made in triplicate.

Solid state spectra were obtained in an Agilent Cary 100 Bio Spectrophotometer and ACB, G9821A at 25 °C. Sample was well ground and placed in a plastic film, and then, measurements were carried out in 200–900 nm spectral window.

#### 3.2.4. EPR

EPR spectra data were collected in a JEOL JES-TE300 (Akishima, Japan) at the X-band with a 9 GHz frequency in a quartz tube at −196 °C with frozen 2.0 mM samples of Cu(II) complexes in MeOH.

#### 3.2.5. Conductivity

Conductivity data were determined with a HI2314 Hanna Instruments equipment (Smithfield, RI, USA) in water. KCl 0.0100 M sample was prepared with ultrapure water, verifying 1412 µS/cm at 25 °C. Samples were prepared at 1 mM concentration, 5 mL were taken and conductivity was measured, and electrodes were washed between measurements with abundant deionized water and dried. Measurements were made in triplicate.

### 3.3. Synthesis of the Copper (II) Complexes 3 and 4

Cu(II) complexes were synthesized as described in patents (Figure 2) [86,87]. In a round-bottom flask equipped with magnetic stirrer was placed 0.182 g (0.96 mmol) of copper(II) nitrate monohydrate, which was dissolved in 50 mL methanol HPLC grade and stirred for 30 min. In a second round-bottom flask was placed 0.173 g (0.96 mmol) of 1,10-phenanthroline dissolved in 25 mL methanol HPLC grade with stirring. In a third round-bottom flask was placed 0.100 g (0.96 mmol) of (*rac*)-3-amino-2-methylpropanoic acid (**1**) or 0.160 g (0.96 mmol) of (*rac*)-3-amino-2-phenylpropanoic acid (**2**) and 0.038 g (0.96 mmol) of sodium hydroxide dissolved in 20 mL of methanol HPLC grade and 5 mL of water. After 30 min, the mixture contained in flask 1 was slowly added to flask 2 and then stirred for 2 h. Following this, flask 3 was added drop by drop, with continuous stirring, for 1 h. Finally, the resulting solution was filtered under vacuum and protected from light over 15 days. The resulting precipitate was recrystallized from a mixture 2:8 MeOH/Et_2_O or EtOH/Et_2_O up to 5 times.

[Cu(Phen)(β^2^-Me-Ala]NO_3_ (**3**). Yield 79%. IR (ATR, cm^−1^): ν = 3227, 3137 NH/OH, 1567 C=O, 1511 C=N, 1335 $NO_{3}^{-}$, 1234 C–O. UV-Vis (H_2_O): λ_max_ = 272 nm, π→π*; λ_max_ = 697 nm d_xz_, d_yz_→dx^2^−y^2^. ε = 30.3 M^−1^ cm^−1^ (λ_max_ = 697 nm). Magnetic susceptibility: µ_eff_ = 2.07 BM. Molar conductivity: Λ_m_(H_2_O, 23.5 °C) = 124.25 S·cm^2^/mol. H.R. M.S. (ESI^+^): calculated: [C_16_H_16_CuN_3_O_2_]^+^ = (345.0538); found: [C_21_H_18_CuN_3_O_2_]^+^ = (345.0522 *m*/*z*).

[Cu(Phen)(β^2^-Ph-Ala)]NO_3_ (**4**). Yield 82%. IR (ATR, cm^−1^): ν = 3237, 3139 NH/OH, 1568 C=O, 1519 C=N, 1326 $NO_{3}^{-}$, 1224 C–O. UV-Vis (H_2_O): λ_max_ = 272 nm, π→π*; λ_max_ = 652 nm, d_xz_, d_yz_→dx^2^−y^2^; ε = 52.3 M^−1^ cm^−1^(λ_max_ = 652 nm). Magnetic susceptibility: µ_eff_ = 1.55 BM. Molar conductivity: Λ_m_(H_2_O, 23.6 °C) = 129.41 S·cm^2^/mol. H.R. M.S. (ESI^+^): calculated: [C_21_H_18_CuN_3_O_2_]^+^ = (407.0695); found: [C_21_H_18_CuN_3_O_2_]^+^ = (407.0676).

### 3.4. Hirshfeld Surface

Hirshfeld surface maps for **3** and **4** were calculated using the Crystallographic Information Files (CIFs) with Crystal Explorer 21.5 software [88]. Hirshfeld surfaces (d_norm_) were mapped in −0.6 to 2.6 range.

### 3.5. Cyclic Voltametric Measurements

Half-wave potential values (E_1/2_) were determined by cyclic voltammetry technique. Measurements were carried out in a three-electrode cell: a 4 mm glassy carbon electrode, a gold electrode and a silver wire as working electrode, counter electrode and auxiliary electrode of the pseudo reference (Fc/Fc^+^ redox couple, according to IUPAC recommendations). Then, 2 mL of a 2 nM solution of the Cu(II) complexes was added in a 10 mL cell, and the cell was filled with a 50 mM solution of NaNO_3_ in dimethylformamide as the supporting electrolyte. Before measurements, solutions were deoxygenated by bubbling nitrogen (99.999% purity). Cyclic voltammograms were recorded in reductive direction at 10, 20, 50, 100, 200, 500, and 1000 mVs^−1^.

### 3.6. Single-Crystal X-Ray Diffraction Analysis

Suitable crystals for X-ray diffraction were isolated from mother liquor solutions and collected at 297.65 K for complex **3** and 99.98 K for complex **4** on Agilent Technologies SuperNova and Bruker D8 APEX2 diffractometers equipped with area detector using Mo-*Kα* radiation. Structures were solved with OLEX2 v1.5 [89] package using SHELXL 2019/3 [90].

### 3.7. Effective Magnetic Moment Measurements

From an empty tared glass tube (*m*_0_) was obtained its magnetic susceptibility (*R*_0_) in a Sherwood Scientific MK-1 Magnetic Susceptibility Balance (Cambridge, UK). Samples were ground until a fine powder was obtained and then were packed into the glass tube at 300 K and weighed (*m*). Magnetic susceptibility (*R*) was measured for samples. The effective magnetic moment (µ_eff_) was calculated according to Equations (1)–(4).(1)$${\;\;X}_{g}=\frac{C*h\left(R-R_{0}\right)}{10^{9}\left(m-m_{0}\right)}$$(2)$${\;X}_{M}=X_{g}\left(MW\right)\;$$(3)$$X_{corr}=X_{m}-\sum \left(diamagnetic\;corrections\right)\;$$(4)$${\;µ}_{eff}=2.84\sqrt{\left(X_{corr}*T\right)}\;$$
where C is the magnetic susceptibility balance constant (1.00426); h is height of the glass tube (cm); MW is the molecular weight (g/mol), and T is temperature (K). Diamagnetic corrections were determined by using tables reported by Berry and Bain [91].

### 3.8. Growth Inhibition (IC_50_ and Selectivity Index)

MCF-7, A549, HEK293, and HaCaT cell lines were maintained at not less than 90% viability employing standard methods at 5% CO_2_ and 37 °C in a cell incubator until used. Before experiments, 1 × 10^4^ cells from each cell line were placed in 96-well microplates containing advanced Dulbecco’s modified eagle medium (DMEM) supplemented with 5% fetal bovine serum (FBS) and 4 mM glutamine solution. Cells were incubated over 24 h at 37 °C and 5% CO_2_ to allow adherence and obtain 95% confluency.

After 24 h, the cell medium was removed and the cells were washed with 100 µL of PBS. Testing compounds dissolved in fresh medium at 0.1, 0.2, 0.4, 0.8, 10, and 20 µM concentrations were added to cells and incubated over 24 h to observe cell growth inhibition. CasII-Gly was used as control, employing the same concentrations. The following concentrations were used to determine the IC_50_ of Cu(NO_3_)_2_: 5, 10, 25, 50, and 75 µM. The previously calculated IC_50_ value for cisplatin for the cell lines evaluated was 25 µM. After incubation time, 10 µL of MTS was added and samples were again incubated over 3 h at 37 °C. MTS activity was calculated by measuring absorbance at 490 nm in ELISA equipment. Data are expressed as percentage of viability; %viability = 100%·[T/U], where T is the viability of treated cells while U is the viability of untreated cells. Growth inhibition data are expressed as IC_50_, which is the concentration needed to inhibit the growth process to half of the maximum. For every cell line were obtained data from three independent experiments.

The Selectivity Index (SI) was calculated by comparing the IC_50_ of complexes **3** and **4** in normal cells (HaCaT) and tumor cells (A549 or MCF-7) [92] using Formula (5):(5)$$SI=\frac{{IC}_{50}(HaCaT)}{{IC}_{50}(A549\;or\;MCF\text{-}7)}\;$$

#### 3.8.1. Staining with Fluorescent Dyes Acridine Orange (AO) and Propidium Iodide (PI)

The purpose of using acridine orange (AO) and propidium iodide (PI) is to take advantage of their introduction of viable cells (AO) and non-viable cells (AO/PI) by their intercalating ability in DNA of cells, which by excitation with a wavelength of 400 nm emit radiation of lower energy and longer wavelength (green and red light, respectively). This technique allows differentiating the distinct states in which a cell is found; AO is able to cross the cell membrane regardless of its membrane integrity, so the compound stains all cells, emitting green light and, in some cases, orange if it binds to RNA or where there is high lysosomal activity [93]. PI penetrates the cell when its membrane is damaged by intercalating with higher affinity to DNA and emits red color, eclipsing the intensity caused by AO staining. Briefly, the protocol used for this technique was: $0.5\times 10^{6}$ cells/mL of A549, MCF-7, HEK293, and HaCaT cells were seeded in 24-well plates on round glass coverslips. All cell cultures were treated with complexes **3** and **4** with the IC_50_ calculated previously [94]. At 12 h, the coverslips were removed from the wells. Later, 5 μL of a 1:1 mixture of the dyes (50 μg/mL of AO and 100 μg/mL of PI) was added to 5 μL of sample and coverslips were flipped and placed on a slide for observation. Subsequently, the sample was observed with TRITC (red channel) filters for propidium iodide and FITCI (green channel) for acridine orange with an epifluorescence microscope (Nikon Eclipse E400, Kyoto, Japan).

#### 3.8.2. Nuclei Staining with Hoechst 33258

Hoechst dyes are excited by UV light (~360 nm) and emit a broad spectrum of blue light with a maximum at 460 nm. Upon binding to DNA, they increase their fluorescence 30-fold, which ensures a good signal-to-noise ratio. This increase in fluorescence is due to the suppression of rotational relaxation and reduced hydration upon binding to DNA. These dyes are not intercalating; they bind to the minor groove of the DNA in the A-T rich regions. Hoechst dyes are often used to distinguish condensed pyknotic nuclei in apoptotic cells [95]. Briefly, the Hoechst dye stock solution was prepared by dissolving the contents of one vial (100 mg) in 10 mL of deionized water to create a 10 mg/mL (16.23 mM) solution. Then, $5\times 10^{5}$ cells/mL of A549, MCF-7, HEK293, and HaCaT cells were seeded in 24-well plates on round glass coverslips. Cell cultures were treated with the complexes **3** and **4** using the IC_50_ calculated previously. At 12 h, the coverslips were removed from the wells. Cells were fixed by adding 200 μL of 4% paraformaldehyde. Later, cells were incubated at 37 °C for 30 min and washed 2 times with PBS. Then, 200 μL Hoechst 33258 (1:1000) was added to the sample and protected from light following incubation at room temperature for 30 min. Subsequently, they were washed 2 times with PBS. Finally, the sample was observed with DAPI (blue channel) filters with an epifluorescence microscope (Nikon Eclipse E400).

#### 3.8.3. DNA Fragmentation Assay

DNA fragmentation is a characteristic process of apoptosis. During apoptosis, cell DNA is broken into fragments of specific sizes (multiples of 180–200 bp) due to the action of nucleases activated by effector caspases (caspase-3, caspase-7, and caspase-6) [96]. To visualize DNA fragmentation on agarose gels, cultures of MCF-7, A549, HEK293, and HaCaT cells treated with complexes **3** and **4** were harvested under the same conditions as the acridine orange/propidium iodide assay and their nuclei stained. Cell cultures were treated with complexes **3** and **4** using the IC_50_ calculated previously. Cultures were harvested and centrifuged at 12,000 rpm; pellets were washed twice with PBS and resuspended in lysis buffer (10 mM Tris-HCl (pH 7.4), 5 mM EDTA, 0.2% Triton X-100) with RNase and incubated for 1 h at 37 °C. Proteinase K was then added to the lysis buffer. Proteinase K was then added and incubated for 2 h at 55 °C. Extraction was performed with phenol:chloroform:isoamyl. To precipitate DNA, 96% ethanol, 10 M ammonium acetate, and glycogen were added. Finally, the samples were resuspended in 1X Buffer TE and stored at −20 °C. DNA was electrophoretically separated on a 2% agarose gel with ethidium bromide and visualized by UV transillumination.

#### 3.8.4. Apoptosis Detection with Annexin V FITC and Propidium Iodide

Annexin V detects apoptosis by binding to phosphatidylserine, which during this process is exposed in the outer layer of the cell membrane. Its conjugation with fluorochromes such as FITC allows visualization of this binding by epifluorescence microscopy in the green channel [97]. The detection of apoptosis was determined with the annexin V kit conjugated with the fluorophore FITC (Thermo, cat. 88-8005-72) under the following protocol: 0.5 × 10^6^ cells/mL of A549, MCF-7, HEK293, and HaCaT cells were seeded in 24-well plates on round glass coverslips treated with L-polylysine. Cell cultures were treated with complexes **3** and **4**, cisplatin was used as apoptosis control, and the concentrations used correspond to the IC_50_s previously calculated in Table 5. Cultures were incubated for 12 h under standard culture conditions. Subsequently, the cells were washed with 500 μL of cold PBS and incubated for 5 min in a 1X solution of annexin-binding buffer. Next, 15 μL of annexin V conjugate and 5 μL of propidium iodide (PI) were added, and the mixture was incubated over 15 min at room temperature. After washing with 1X buffer, the cells were observed using an epifluorescence microscope (Nikon Eclipse E400). The red channel (TRITC) identified necrosis due to PI intercalation, while the green channel (FITC) detected apoptosis through binding of the phosphatidylserine–annexin V complex.

## 4. Conclusions

Two novel Cu(II) complexes containing 1,10-phenanthroline as the primary ligand and β^2^-aminoacidates as secondary ligands were synthesized with good yields. X-ray crystallographic analysis and cyclic voltametric characterization of both complexes showed a square-based pyramidal geometry and a quasi-reversible behavior. Biological assays showed an excellent inhibition activity on the MCF-7 and A549 tumor cell lines, being even more potent than cisplatin. Induction of cell death by apoptosis in cancerous cells is one of the main focuses in the search for new chemotherapeutic compounds, so the characterization of cell death by techniques such as those presented in this work is rather relevant. It is noteworthy that specificity is not observed for compounds **3** and **4**; they impact the survival of both tumor cells (A549 and MCF-7) and non-immortalized cells (HEK293 and HaCaT). These results are relevant in the pharmaceutical context as they provide a promising starting point for structural optimization. In particular, the development of nanoparticle-based formulations could significantly improve specificity by targeting the compounds to tumor cells, thus reducing their impact on normal cells. Likewise, the design of prodrugs based on the structures of complexes **3** and **4** could allow controlled activation in the tumor micro-environment, improving both efficacy and the safety profile. The ability of these compounds to induce apoptosis in cancer cells underlines their potential as candidates for chemotherapy, highlighting the importance of these strategies in the search for new anticancer drugs.

## Data Availability

There is no server where data can be saved for consultation but, if needed, data can be shared by contacting jaime@uaem.mx.

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
