# Peer review of "Synthesis of (1,10-Phenanthroline-κ2N,N′)(β2-Methyl- and β2-PhenylAlaninate-κ2N,O)Copper(II) Nitrate Complexes and Their Antiproliferative Activity on MCF-7 and A549 Cancer Cell Lines"

_molecules, 2025, doi:10.3390/molecules30030634_

Round 1

Reviewer 1 Report (New Reviewer)

Comments and Suggestions for Authors

The presented work describes the synthesis and precise identification of two new copper(II) complexes and the study of their potential antiproliferative activity against selected cancer cell lines. The work is well planned and organized, references are extensive and well selected.

Minor comments mainly concern technical issues and the presentation of results.

1.       Scheme 1 describes the synthesis of ligands, while the synthesis of complexes 3-4 (Scheme 2) is presented only on page 20. Both diagrams should be included in the Results and discussion section.

2.       The spatial arrangement of some figures requires improvement. Figure 2 is cut off, making it difficult to interpret. Signature CCDD also is cut off and moved to the margin. Figure 4 is also cut off. Figure 5 - legend text is truncated.

3.       The origin of the cell lines used in the research (A549, MCF-7, HaCAT, HEK293) should be explained in the first place they appear in the article (it is only in the methodology).

4.       Table 5 - please provide information on the number of repetitions of the experiments. Do the presented deviation values ​​refer to SEM? A value of 25 µM for cisplatin was calculated earlier. Is there a link to this work?

5.       What is the solubility of the complexes obtained? For biological studies, were they dissolved directly in the medium or first in another solvent, e.g. DMSO?

6.       Reference 79 gives the names, not the surnames, of the authors of the article.

Author Response

Reviewer 2 Report (New Reviewer)

Comments and Suggestions for Authors

The manuscript by Escalante and co-authors reports the synthesis of two new Cu(II) complexes from the Casiopeínas® family of compounds. The novel Cu(II) complexes were characterized by various techniques. The complexes were also investigated in vitro as potential anti-tumor drugs. The manuscript is very interesting and could contribute to the current theme of developing metal-based drugs, but it needs some major revisions before acceptance. I hope that the authors will improve the manuscript with a new revision.

Major points

1. The investigation of interactions with DNA and HAS could be beneficial for understanding the nature of the bonds between the Cu(II) complexes and biomolecules. The authors could use molecular docking analysis.

2. The index selectivity (SI) should be determined to indicate the specificity of the complexes towards cancer cells.

3. IC50 results for the copper(II) complexes need to be redetermined after 48 or 72 hours.

4. The conclusion should be more scientific, reflecting the scientific findings.

Minor points

1. The results of elemental analysis should be included in the experimental section.

2. Figures 2 and 4 should be placed in the correct positions.

3. Every paragraph should be indented.

4. In Table 5, it should be clarified that the results were obtained after 24 hours.

5. Consistent units should be used; sometimes temperature is given in K, and other times in °C.

6. Line 517 should be written as: "Cytotoxic activity and effects of complexes 3 and 4 on the induction of apoptosis are presented in Figure 14..."

7. The abbreviations for cancer cells should be defined at their first appearance.

Round 2

Reviewer 2 Report (New Reviewer)

Comments and Suggestions for Authors

The authors have improved their work, and it is completely acceptable in its current form.

This manuscript is a resubmission of an earlier submission. The following is a list of the peer review reports and author responses from that submission.

Round 1

Reviewer 1 Report

Comments and Suggestions for Authors

A manuscript by Escalante and coworkers describes the preparation and characterization of two Cu(II) complexes with a mixed ligand system: phenanthroline/β2-aminoacidate, and their use as potential anticancer compounds.

Although the topic is generally interesting, the present paper has several issues that, in our opinion, must be addressed before it can be published. Below, we will comment on the three aspects of the work: (i) conceptual; (ii) technical; and (iii) formatting.

(i) Conceptual and Critical

  1. The most serious concern is the unsubstantiated claims regarding the structures of the studied complexes in solution versus those revealed by X-ray crystallography. There are no experiments in the paper confirming that the stoichiometry of the complexes remains the same in solution as in the solid state. The authors did not discuss speciation in the solution and do not know which species are responsible for the biological activity.
  2. Numerous control experiments are missing, which are especially critical for the in vitro studies on cell cultures (no data presented for cell viability/inhibition without the complexes in question, as well as with only Cu(II), only with phenanthroline, etc.). This also applies to the spectral studies, namely UV-Vis and EPR (only Cu(II) and Cu(II)Phen as controls in both methods).
  3. There is a critical issue with the IR measurements for Cu(II) compounds in KBr pellets. It is well known that KBr is often unsuitable due to reactivity with Cu(II) during grinding in sample preparation.

(ii) Technical

  1. Numerous experiments are poorly described: concentrations, sample preparations, etc.
  2. UV-Vis spectroscopy: i. Sample preparation, working concentration, and cuvette path length are missing; for solid-state spectra, experimental conditions are missing. ii. Control experiments (only Cu2+, Cu2+ with Phen, etc.) are missing. iii. Different solvents should be tested to see if there is any difference (e.g., methanol, THF). iv. Measure absorption at various concentrations; even if the absorption below 250 nm will be saturated, it is important to see whether there will be changes in the absorption band of Cu(II) transitions. v. Otherwise, use a cuvette with a shorter path length (e.g., 1 mm). vi. Figure 3 misses y-axis values (a.u.). vii. Figure 3: why is absorption at 6 a.u.? It is beyond the usual linearity range. Or is it just a superposition of two spectra in one figure? If so, it should not be done this way. viii. Indicate in the caption that the inset represents a zoomed-in region; indicate the concentration and solvent. ix. Instead of a structure of the complex determined by X-ray, indicate just the number of compounds 3 and 4. At such concentrations in water, no ethanol is likely coordinated to the copper ion. For solid-state spectra, you may leave it as it is. x. Line 156: the comment on stability is unsubstantiated. Changes in such complexes will happen almost instantaneously in water, right after dissolution (dissociation, ligand exchange, new speciation equilibrium).
  3. EPR spectroscopy: i. The concentration of the sample is missing. ii. Control experiments are missing. iii. For EPR spectra, it is not necessary to have a y-axis as it is the first derivative of the absorption intensity; values of the x-axis are missing. iv. We suggest adding a spin-spin coupling pattern to the EPR spectra with the indication of a coupling constant A. v. In solution, one can estimate the coordination pattern (e.g., CuN4 vs CuN3O, CuN2O2, etc.) by considering the relationship between the constants g and A (see, for example, Bennett, B.; Kowalski, J. M. EPR Methods for Biological Cu(II): L-Band CW and NARS. In Methods in Enzymology; 2015; pp 341–361. https://doi.org/10.1016/bs.mie.2015.06.030). Please comment on this in the discussion of your results.
  4. CV measurements: i. Lines 449-450: conditions are not clear.

(iii) Formatting

  1. Please check the ACS Style Guide or other scientific style guides.
  2. Line 36: the use of commas for delimiting thousands is confusing.
  3. There is poor consistency throughout the text (see line 109: 3200 - 3600 cm-1 vs line 112: 1224 – 1227 cm-1) and in numerous other places (lines 185, 406, 409).
  4. Line 114: NO3- - the minus sign is incorrectly positioned; this issue occurs in many other places as well.
  5. The synthesis of complexes should be presented in section 2.1, as the reader should see the structures of the discussed compounds at the beginning of the paper.
  6. SI is not properly cited in the main text.
  7. SI contains material in Spanish.

Reviewer 2 Report

Comments and Suggestions for Authors

The current manuscript presents two copper complexes which have been prepared, well characterised, including XRD and EPR spectroscopy, and these then subjected to a suite of biological testing. As it stands the manuscript cannot be accepted for publication due to two main concerns which are listed below:

1. Only two complexes are prepared and studied. These in and of themselves are not really very interesting, and it would be far more interesting if a much larger number of homologues were prepared and then tested. This way, SAR could be delineated. I would advise the authors to prepare several more related complexes of this type and test them to see what drives the biological activity. Two new complexes are not enough to warrant a full paper, in my opinion. 

2. Very important: Hydrolysis studies need to be done, to determine what the fate is of the complexes in biological media. This can be done using a PBS (or similar) buffer, at 37 C, and studying it by a combination of UV/Vis and mass spec. It is very likely that the pre-drug undergoes hydrolysis over time and that the species responsible for biological activity is not that of the starting complex. This needs to be studied. 

If the authors substantially increase the number of new compounds related to these reported, expand the scope of testing, and include hydrolysis investigations, the work may be more interesting to the readership of Molecules. I encourage the authors to include this and the new results and re-submit at a later date.

Reviewer 3 Report

Comments and Suggestions for Authors

The paper by Liu et al. reports the two novel copper(II) complexes containing 1,10-phenanthroline as primary ligand and β2-aminoacidates as secondary ligands. X-Ray crystallographic analysis and cyclic voltametric characterization of both complexes showed a square-based pyramidal geometry and a quasi-reversible behavior. Both complexes showed better antitumor activity than cisplatin in cancer cells, but they were also highly toxic to normal cells. In my opinion, this work can be considered for publication in “Molecules” after revision.

1. The authors can test the spectral changes of the complex over time to judge whether the complex will undergo hydrolysis.

2. Reactive oxygen species are important factors in cell death induced by metal complexes. Intracellular reactive oxygen species levels shoule be detected to evaluate the cell toxicity casued by complexes.

3. One of the graphs in Figure 7 is missing the scale bar.

4. Annexin V-FITC/PI staining is the most standard assay for the evaluation of apoptosis, and the authors may consider performing it.

5. A minor suggestion, some recent references to the treatment of cancer through copper complexes, should be cited in this revision, like Dalton Trans., 2022, 51, 16574-16586; J. Inorg. Biochem., 2023, 246, 112299.

Reviewer 4 Report

Comments and Suggestions for Authors

Comments on the Quality of English Language

Round 2

Reviewer 1 Report

Comments and Suggestions for Authors

Dear Colleagues, dear Editor,

From the response of the authors, I am not sure I was clear enough in my first review regarding the critical points in the manuscript, as their response on Point 1, unfortunately, completely misses it.

The crystal structure of the studied complexes is not under critique. This part of the study is of high quality and is quite solid.

A very weak point of their interpretation of the results is: to consider, that at concentrations as low as 105 M (for UV-Vis studies), or even 109 M (for electrochemical studies) the stoichiometry of the studied complexes remains the same as in the solid state. – That is unsubstantiated.

Complex equilibria exist in solutions of metal complexes. The lower the concentration of the complex, the higher the competition with the molecules of solvent.

For instance, [CuL1L2L3]+ under dissolution (in water for example), may (and undoubtedly will) undergo a series of transformations giving rise to multiple species, such as all possible forms with water in the first coordination sphere [CuLm(H2O)n]x+; species with homoleptic ligand systems such as [Cu(L1)2]2+, [Cu(L2)2], etc. This process is also pH-dependent.

In addition, quoting the previous works on this topic, to support the interpretation of spectroscopic data is of little help, if the interpretation of spectroscopic data in those works was done in a similar way.

This is what is concerning regarding the methodology and science part.

The second point of concern is regarding scientific rigor.

I will quote here only 3 points:

1)      IR studies. As was indicated in my first review, it is better to avoid measuring the IR spectra of Cu(II) compounds in KBr due to their cross-reactivity. The authors have done a good job in re-measuring the spectra in ATR mode, which is more suitable.

Yet, in the characterization part (e.g., line 669), KBr is still mentioned.

If one compares the reported values with the IR spectrum in the SI (e.g., Figure S6), one can notice that they are all wrong (probably because the authors forgot to report the new ATR-FTIR data). It happens, of course…

Then, the authors respond that spectra obtained with ATR and in KBr are comparable. They are comparable only if one does not look at the frequencies. If one does, one can see significant shifts of certain bands (ca. 20 cm1 for C=O band), suggesting constitutional changes in the structure of studied samples on grinding with KBr vs ATR-mode.

2)      UV-Vis studies. The only reliable data regarding the UV-Vis of the actual reported compounds is Figure 8 (C and D), which reports UV-Vis absorption in the solid state.

All the solution UV-Vis studies are inconclusive. Even in terms of representation. For instance, in the SI, Figures S10-S13 report UV-Vis spectra at higher concentrations. Beyond absorption of 3 a.u. they are, unfortunately, above the linear range of the instrument (saturation of the signal), thus no useful information can be extracted. The authors should have tried to use a cuvette with a shorter path (1 mm, 0.1 mm).

Then, the authors measured the stability of the complexes. First, the presentation is suboptimal; one should overlay T0, T24, and T48 for comparison on the same figure. Furthermore, see for instance figures s16 and s17. Absorption at ca. 200 nm in Fig. S16 is slightly above 2.5 a.u., while in Fig. S17 it is at ca. 2.3 a.u level… Either it does not prove the stability, and thus the claims, or something is wrong. And that is without considering that in terms of dissociation and speciation, only seconds are needed in water, and not 24 h. Thus, this experiment is of a low value.

3)      EPR studies. gz is determined wrongly, completely... Moreover, I would not be so confident as to say that gx and gy components are equal. I would strongly suggest consulting with a specialist in Cu EPR.

With this in hand, unfortunately, I cannot recommend publishing this paper in such a form. It should be reworked and improved, with rigor and patience. The aim is to do good-quality science.

Reviewer 2 Report

Comments and Suggestions for Authors

My initial comments have not been addressed:

More than two complexes are required for the manuscript to form a coherent picture. It is typical in biological studies of this type to present several (not just two) related complexes in one paper, and present a full picture. This has not been done. I would recommend that authors isolate more related complexes, test them, and derive a structure activity relationship. This will be far more meaningful than just having two data points, or making more complexes and publishing them in separate publications. 

The fate of the complexes in biological media has also not been addressed. One needs to perform UV Vis studies in buffer solution, perform kinetic analysis, and see what the fate of the molecules are in biological media (are there changes? is water coordinating ? how fast or slow ?). This has still not been done. Alternatively, one can stir the complexes in PBS buffer and perform Mass spec. Obviously the complexes will react somehow in these systems, and this needs to be studied. 

In light of this, and the fact that these two concerns were not dealt with, I cannot recommend acceptance of the paper.

Reviewer 3 Report

Comments and Suggestions for Authors

The authors revised the most part of their manuscript according to my suggestions. I have no claim in the revised paper.

Reviewer 4 Report

Comments and Suggestions for Authors

Comments on the Quality of English Language

Minor issues.